# Assessment of Oxidative Stress Indices and Total Phenolics Concentrations in Obese Adult Women—The Effect of Training with Supplemental Oxygen: A Randomized Controlled Trial

**DOI:** 10.3390/nu15010241

**Published:** 2023-01-03

**Authors:** Katarzyna Domaszewska, Agnieszka Zawada, Radosław Palutka, Tomasz Podgórski, Aldona Juchacz

**Affiliations:** 1Department of Physiology and Biochemistry, Poznan University of Physical Education, 61-871 Poznań, Poland; 2Department of Gastroenterology, Dietetics and Internal Medicine, Poznan University of Medical Sciences, 61-701 Poznań, Poland; 3Eugenia and Janusz Zeyland Greater Poland Pulmonology and Thoracic Surgery Centre, 60-569 Poznań, Poland

**Keywords:** TBARS, *C*-reactive protein, reactive oxygen species, acid–base equilibrium

## Abstract

Background: The aim of this study was to determine the effect of using an oxygen-enriched breathing mixture during controlled physical training on blood oxidative stress parameters and total phenolics (TP) concentrations in obese adult women. Methods: A prospective randomized controlled trial study included 60 women aged 19–68 with BMIs greater than 30 kg/m^2^. Patients were randomly assigned to the study group (*n* = 30), which received additional intervention in supplementing the breathing mixture with oxygen at the flow of 6 L/min during training sessions, and the control group (*n* = 30). At the beginning and at the end of the study, anthropometric assessments (height and weight and BMI) and blood tests (CRP, FRAP, TBARS, TP, BAC, and La) were performed. For each patient, an individual endurance training plan was established on a cycloergometer, including 12 training units, based on a cardiopulmonary exercise test (CPET). Results: A decrease in blood TBARS concentration was observed in each study group. For the control group, the change was more remarkable, and the difference between the groups was significant at (*p* < 0.05; ES: 0.583). Training with the oxygen breathing mixture increased blood concentrations of TP, while a decrease in TP in blood was observed in the group without oxygen supplementation during physical training. The difference in the responses between the groups was significant at (*p* < 0.05; ES: 0.657) Conclusions: Increasing the concentration of oxygen in the respiratory mixture under conditions of increased exercise was shown to be safe because it did not exacerbate oxidative stress in the obese group.

## 1. Introduction

The processes of proper oxygen supply to tissues and cells, the control of mediators of the inflammatory process, and balance in the oxidation–antioxidation system occur simultaneously in living organisms in close interdependence with each other. Appropriate balance between them is a prerequisite for health and wellbeing. The dysfunction of any element entails changes in the others, leading to the development of a wide variety of diseases ranging from cardiovascular conditions, metabolic disorders, degenerative processes, neurodegenerative processes, and finally, cancer [1,2]. Oxidative stress refers to an imbalance between the production and/or accumulation of reactive oxygen species (ROS) in cells and the body’s ability to neutralize their harmful actions. The defense system is a set of endo- and exogenous substances (mainly enzymes, but also small-molecule compounds such as glutathione, uric acid, and coenzyme Q) which forms the antioxidant potential of the plasma [3]. Reactive oxygen species—superoxide anion radical, hydrogen peroxide, hydroxyl radical, and singlet oxygen—are physiologically produced during metabolic processes of the body and are controlled by the antioxidant system under healthy conditions [2,4,5]. Under unfavorable conditions—inflammation and hypoxia, as well as under the influence of environmental factors, including certain drugs, agents used in the food industry or in ionization or ultraviolet radiation, and environmental pollution, especially smog fog—they are overproduced. Low physiological levels of ROS in a cell control the processes of protein phosphorylation, the activation of transcription factors, and apoptosis, which are desirable activities [6]. In contrast, the uncontrolled growth of ROS damages elementary cellular structures such as lipids, which are indispensable parts of cell membranes or protein structures [7]. In such cases, endoplasmic reticulum hypertrophy, mitochondrial atrophy, cell membrane degradation, and cell death are observed. Excess free radicals can also damage nucleic acids, destabilizing DNA, leading to mutagenesis, and causing the loss of epigenetic information in some cases through the disruption of methylation processes in gene promoters [8]. Striving for balance, the body produces defense mechanisms which include antioxidant enzymes (glutathione peroxidase, glutathione reductase, catalase, and superoxide dismutase) and low-molecular-weight non-enzymatic components (selenium, ascorbic acid, vitamin A and E, glutathione, and total phenolics) which nullify the effects of ROS [9,10].

Obesity is a condition which in itself induces oxidative stress, and the coexistence of hypoxemia and chronic inflammation further exacerbates its effect. De Ferranti and Mozaffarian demonstrated an increase in malondialdehyde (MDA) levels in the fatty tissue of obese individuals, which is manifested by the presence of lipid peroxidation processes of cell membranes in these patients [11]. The mechanism of positive feedback of oxidative stress and obesity still requires in-depth studies. In recent years, the effect of adipokines on ROS production has been demonstrated [12]. Additionally, circulating free fatty acids by uncoupling the respiratory chain in mitochondria can generate the production of higher amounts of ROS [13].

The aim of treating obesity is to compensate for energy balance through diet and exercise. Only in this way can we reduce metabolic inflammation and oxidative stress and reduce the burden that excess weight places on many systems, including the cardiovascular, respiratory, and osteoarticular ones. Currently, there are three main methods of treating obesity: the method of changing dietary habits connected with applying physical exercise, the use of pharmacotherapy, and surgical treatment. The last two methods of treatment carry a high risk of complications or side effects, and the results are not always satisfactory.

The aim of this study was to determine the effect of using an oxygen-enriched breathing mixture during controlled physical training on the total phenolics concentration in plasma and oxidative stress indices in obese female patients. We hypothesized that the controlled use of oxygen therapy as a supportive measure in the treatment of obesity would not cause oxidative stress in the examined women.

## 2. Materials and Methods

A randomized controlled trial was conducted. Sixty-two women were initially enrolled in the experimental group and randomly assigned to the study group (with additional intervention) and the control group (without additional intervention), after informed written consent was obtained from these subjects. The women were recruited for the study from patients of the Metabolic Clinic. Two participants did not complete the therapeutic process due to health conditions that precluded participation. Randomization was conducted as a simple assignment (Excel; Microsoft, 2016 software), and the person performing it was not involved in the research process.

All the study subjects were Caucasian and from the Wielkopolska region (Poland). Before entering the project, all the participants were obliged to strictly follow dietary recommendations and the diet individually prepared for use in the study by the included dietician. Those enrolled in the study during the project were asked not to perform additional physical activity beyond that carried out as part of the program. The examined women had not taken a part in a similar research project before.

The investigated women were between 19 and 68 years of age, with a Body Mass Index (BMI) > 30 kg/m^2^. Eligibility for participation in the study was determined after prior medical consultation. The inclusion criteria for the study included the absence of comorbidities that prevented physical effort. The criteria for excluding a patient from the study were the presence of at least one of the following factors: positive electrocardiographic exercise test result; adherence to a vegetarian or another alternative diet; historical or active cancer (ongoing radiation/chemotherapy); patients with liver disease (ALT > 3x normal limit), except patients with hepatic steatosis; patients with chronic kidney disease eGFR < 30 mL/1.73 m^2^/min; patients with active inflammation CRP > 5 mg/dL; patients with unstable ischemic heart disease; patients who had undergone myocardial vascularization or pacemaker implantation; patients who had undergone ischemic or hemorrhagic stroke (<6 months); patients who had undergone STEMI with drug-eluting stent implantation or nSTEMI (<12 months); patients with inherited metabolic disorders: phenylketonuria or galactosemia; patients with autoimmune diseases; pregnant patients; patients with comorbid psychiatric or eating disorders (anorexia or bulimia); patients undergoing antibiotic or steroid treatment; and patients experiencing active substance abuse. The study was performed at the Department of Gastroenterology, Dietetics and Internal Medicine, the Metabolic Outpatient Clinic of the Clinical Hospital No. 2 in Poznań, and the Pulmonology and Rehabilitation Department for Adults of the Eugenia and Janusz Zeyland Wielkopolska Center for Pulmonology and Thoracic Surgery in Poznan, in cooperation with the Department of Physiology and Biochemistry, Poznan University of Physical Education in Poznań (Poland). The study was conducted following the Declaration of Helsinki and received a positive opinion issued by the Bioethics Committee operating at the Poznan University of Medical Sciences dated 6 April 2017; Number: 429/17.

### 2.1. Basic Anthropometric Measurements

The following measurements were taken twice, at the beginning and at the end of the treatment program: height, weight, and BMI. Height was measured with an anthropometer (accuracy ± 1 mm), and body weight was measured with a digital scale (±100 g) with the use of WPT 60/150 OW medical scales (Radwag^®^, Radom, Poland). The differences between the three body weight and height measurements taken were <1%. BMI was calculated and classified in accordance with the WHO criteria.

### 2.2. Methodology for Determining the Level of Physical Fitness and Determining Individual Training Loads

#### Exercise Test Execution Protocol

Exercise testing occurred between 8:00 and 10:00 am in an air-conditioned exercise testing laboratory, 2 h after eating a light breakfast. The Cardiopulmonary Exercise Test (CPET) test was performed on a cycloergometer: a 3 min warm-up with a load of 25 watts, after which, the load was increased by 10 watts (60 RPM) every 90 s. The test continued until the subject refused or was unable to maintain the set cadence. A physician was present in the laboratory during the exercise test in each case.

### 2.3. Analysis of Gasometric Parameters and Physiological Indicators during the Exercise Test

Throughout the exercise test, expiratory gas pressure, minute ventilation (VE), and heart rate (HR) were monitored continuously using an automated START 2000 M system (MES Sp. z o. o., Kraków, Poland). Heart rate was measured using a Nonin 8500 pulse oximeter (Plymouth, MN, USA). Oxygen uptake (VO_2_) and carbon dioxide output (VCO_2_) were measured using a breath-by-breath method and were averaged over 15 s periods. Before each trial, the system was calibrated according to the manufacturer’s instructions. Ambient conditions were recorded using sensors: temperature, humidity, and atmospheric pressure. The calibration of the gas analyzer was performed using a standard gas mixture of 5% CO_2_ and 16% O_2_ provided by the manufacturer. The ventilatory threshold (VT) was determined via the V-slope method, using computer regression analysis of the slope of the graph of CO_2_ production versus O_2_ intake to determine the point at which the increase in CO_2_ production disproportionately exceeded the increase in O_2_ consumption [14]. The load determined during the study was used to determine each patient’s individual training load.

### 2.4. Analysis of Blood Biochemical Parameters

#### 2.4.1. Preparation of Blood Samples for Analysis

The subjects were informed about the conditions for blood collection, i.e., no intense physical activity in the 24 h before the test and a recent low-fat meal. Blood samples were collected during the morning fasting from the ulnar vein using an S-Monovette syringe (SARSTEDT, Nümbrect, Germany) and centrifuged for 4 min at 1500× *g*, nt 4 °C, using a Universal 320R centrifuge (Hettich Lab Technology, Tuttlingen, Germany) to separate plasma. The samples were cooled and stored at −80 °C using a low-temperature freezer (U410, New Brunswick Scientific, Enfield, CT, USA) until analysis.

#### 2.4.2. Acid–Base Equilibrium Parameters and Lactic Acid Concentration in Capillary Blood

Acid–base equilibrium (BAC) parameters were determined in capillary blood collected from the fingertip using a Cobas b221 critical parameter analyzer (Roche Diagnostics, Indianapolis, IN, USA). The following parameters were statistically analyzed: blood pH, pCO_2_, pO_2_, HCO_3_^−^, and BE. Blood oxygen saturation (SpO_2_) was also determined. The measurement of the lactate concentration in capillary blood was determined using an EDGE Blood Lactate Test Strip, ApexBio (Hsinchu, Taiwan).

#### 2.4.3. Methodology for the Determination of Biochemical Indicators in Venous Blood

The determinations were carried out in the laboratory of the Department of Physiology and Biochemistry of the AWF in Poznań according to the procedure described by Domaszewska et al. [15]. Venous blood plasma concentrations were determined:

The FRAP determination was based on the method of Benzie et al. [16]. The total antioxidant capacity of plasma (FRAP) had the following reference values at rest: 600–1600 μmol/L. It involved a reduction of the Fe^3+^-TPTZ (2,4,6-tripyridyl-s-triazine; Sigma-Aldrich, Gillingham, UK) complex to a blue complex of Fe^2+^-TPTZ. The color intensity of the resulting solution was directly proportional to the antioxidant power of plasma. A total of 10 μL of plasma was diluted with 30 μL of deionized water, followed by adding 300 μL of reactive solution (2,4,6-tripirydylo-s-triazine (TPTZ) + ferric chloride III (FeCl_3_·6H_2_O; POCH, Gliwice, Poland) + acetate buffer (pH = 3.6, reagents from POCH, Gliwice, Poland)). After 6 min of incubation at 37 °C, the absorbance was determined on a multidetector microplate ELISA reader (Synergy 2 SIAFRT, BioTek, Winooski, VT, USA) at λ = 593 nm. The standard curve was established using a stoichiometrically diluted solution of iron sulphate II (FeSO_4_·7H_2_O; POCH, Gliwice, Poland).

The determination of TBARS was based on the method by Ohkawa et al. [17]. The concentration of TBARS in the plasma of healthy people was 1–6 μmol/L. This method involved the condensation of MDA with thiobarbituric acid and the formation of a dye compound with 50 μL of plasma, 50 μL of sodium dodecyl sulphate (SDS; POCH, Gliwice, Poland), 375 mL of 20% acetic acid (POCH, Gliwice, Poland), and 375 mL of 0.8% thiobarbituric acid (Sigma-Aldrich, Gillingham, UK), which was placed in a water bath at 95 °C for 60 min. After incubation, the sample was cooled, and the elution was made into a solution of a dye compound of *n*-butane (POCH, Gliwice, Poland). After centrifugation, the upper layer of the solution was separated, and the measurement was performed on a multi-detector microplate ELISA reader (Synergy 2 SIAFRT, BioTek, Winooski, Vermont, USA) at λ = 532 nm. The standard curve was created from a stoichiometrically diluted solution (1,1,3,3-tetramethoxypropane (TMP; Sigma-Aldrich, St. Louis, MO, USA).

The concentration of phenolic compounds in the blood was determined using the method developed by Singleton and Rossi [18]. The proper resting reference values were assumed to be the concentration of total phenolics, which was equal to 2.8–4.0 g GAE/L. This method exploited the ability to oxidize phenolic groups via the Folin–Ciocalteau reagent. The resulting compounds were converted to a blue complex. The color of the solution was measured using a Marcel Media plus spectrophotometer (Marcel sp. z o.o., Zielonka, Poland) at λ = 765 nm. The standard curve was established using standard solutions of gallic acid (GAE). The concentration of total phenolics was expressed as a GAE equivalent in g/L of plasma.

The CRP protein concentration was measured using the latex immunoprecipitation method with turbidimetric measurement on a Cobas Integra 400 plus analyzer (Roche Diagnostics, Basal, Switzerland). The CRP concentration in the plasma of healthy people was noted as being between 0.02 and 13.5 mg/L [19].

### 2.5. Training Program

Both the study and control groups underwent a controlled endurance training process on a bicycle cycloergometer (Kettler DX1 Pro, Ense, Germany) for 3 weeks. The subjects completed 12 training units under the supervision of a physical therapist and a physician. A single training session was 50 min long and consisted of a 5 min warm-up, 40 min of specific training, and 5 min of unloaded cycling. Until the heart rate returned to pre-exercise values, the patient stayed on the cycloergometer.

Patients assigned to the study group received additional intervention in the form of the supplementation of the breathing mixture with oxygen at the flow of 6 L/min during the training sessions. Oxygen was administered via an intranasal cannula using a Korgiel MTO_2_ dispenser with a humidifier. According to the characteristics of the medicinal product—medical oxygen—its effect disappeared after the end of the administration. This could be proven by the drop in saturation after the oxygen was disconnected. Thus, there was no carryover effect on the next day’s training or final examination. The control group breathed atmospheric air during training.

### 2.6. Statistics

The results of the study are presented in tables as the medians and interquartile ranges (Me (Q1–Q3)). The change in the range of selected parameters (Δ) within the study groups was calculated by subtracting the values in the 2nd term from the values in the 1st term of the study. The normality of distribution was tested using the Shapiro–Wilk test. The significance of differences between the values recorded before and after the completion of the intervention for each study group was performed using Wilcoxon’s non-parametric paired-order test. The significance of the differences between the post-training change in values of the studied parameters between groups was calculated using the U-Mann–Whitney test. The correlation was calculated using Spearman’s rank-sum test. The results were statistically analyzed using Dell Statistica data analysis software (version 13, software.dell.com, Dell Inc., Round Rock, TX, USA). Test results were considered significant starting from a significance level of *p* < 0.05. Effect sizes (ESs) were calculated as the difference between means divided by the combined standard deviation. Using Cohen’s (1988) criteria, an effect size ≥0.20 and <0.50 was considered small, ≥0.50 and <0.80 medium, and ≥0.80 large [20].

## 3. Results

The analysis included 60 women aged 19 to 68 years; 47.00 (32.00–54.00). They were randomly assigned to the study group or the control group. The comparison of baseline anthropometric, spirometric and blood biochemical parameters measured in the study showed no significant differences between the study groups. Only with regard to the total polyphenol content of the blood was its significantly higher concentration in women in the control group found (*p* < 0.0001; ES: 1.125) These values for the study group were below the reference standard range (Table 1). The most common comorbidities in the study and control groups were hypertension (37% vs. 27%), hypothyroidism (14% vs. 8%), glucose intolerance, and diabetes (27% vs. 24%). Respiratory diseases accounted for a small percentage of the reported conditions in both study groups. The patients’ chronic treatment included β-blockers, hypotensive drugs, hypoglycemic drugs, normalizing thyroid function, and lipid metabolism. The patients’ pharmacotherapy was not modified during the project.

Table 2 shows the post-training analysis of changes in blood gasometric indices and resting lactate levels at each study date for the test and control groups. Training with an oxygen-enriched breathing mixture at a flow rate of 6 L/min resulted in a statistical increase in blood oxygen tension (pO_2_) during the second research period. No such change was observed for the control group. For this group, physical training led to a statistical decrease in blood pH (*p* < 0.05) and a significant increase in resting blood lactate concentration. (*p* = 0.0578). There was no statistical difference in response to the applied physical training in the analysis of changes between the study groups.

The statistical analysis of the results of plasma antioxidant stress indices and CRP concentrations showed no significant post-training change in the two study groups. A decrease in blood TBARS concentration was observed in each of the study groups. For the control group, the change was greater, and the difference between the groups was significant at *p* < 0.05; ES: 0.583. Training with an oxygen-enriched breathing mixture increased blood concentrations of total polyphenols, while a decrease in total polyphenols in blood was observed in the group without oxygen supplementation during physical training. The difference in the responses between the groups was significant at (*p* < 0.05; ES: 0.657). This may suggest that the resulting oxidative stress in the body is a stimulator of beneficial adaptive changes and results in an increase in the body’s oxidative potential.

## 4. Discussion

The mechanism of damage to the LDL fraction of cholesterol is associated with the development of atherosclerosis, a disease that very often interacts with obesity. It was shown that BMI values correlated significantly positively with blood MDA concentrations. The analysis involved obese individuals, so the high BMI value was due to excessively accumulated fat mass in these individuals [21]. Our study failed to show a similar correlation. The subjects in our study design had blood TBARS concentrations at 5.35 ± 1.586 for the study group and 6.31 ± 2.657 mmol/L for the control group at the beginning of the therapeutic process. These values were within the normal upper range of reference levels and did not differ significantly between the two groups (*p* > 0.05). Within each study group, the applied therapeutic action decreased TBARS concentrations during the second research period. In the control group, the post-training decrease was significantly greater, and the difference in response between the groups was statistically significantly different (*p* < 0.05). According to the theory of Dandon et al., weight loss following caloric restriction in the diet reduces the production of reactive oxygen species by leukocytes, while in our study, physical activity led to weight loss and was responsible for the decrease in blood TBARS concentrations [22]. Similar observations were made by Yesiburs et al. and Prazny et al. in their study on the effect of weight loss with pharmacological intervention with orlistat on blood MDA concentrations in normal and overweight subjects [21,23].

The results of our study indicate that control subjects who underwent moderate-intensity physical training had lower levels of lipid peroxidation during the second research period. This somewhat favorable change was accompanied by a greater post-workout decrease in FRAP and TP concentrations in blood during the testing second research period compared to the test group. This smaller decrease in FRAP concentrations in the subjects of the study group, despite an approximately 4-fold decrease in blood TBARS concentrations, can be explained not only by its reduced de novo synthesis or increased hepatic utilization, but also by a change in the concentration of inhibitors that modify the reaction rate of the described biochemical reaction.

In their study, Schröder et al. attempt to explain the mechanism of the change in the concentration of oxidative stress markers under the influence of exercise. They explain this fact by activating an additional pool of antioxidants in the body which act protectively on cell membranes [24]. The main element here is uric acid, the share of which in the total pool of antioxidants increases to more than 60%. On the other hand, other researchers such as Cao and Prior give more importance to thiol compounds, antioxidant enzymes, and vitamins in combating oxidative stress than that shown by Schröder et al. [24,25]. On the basis of the analysis of our results, we showed an increase in the TP concentration in the blood of subjects in the study group during the second research period (*p* = 0.0889), in contrast to a decrease in the concentration of this indicator in the control group (*p* = 0.8445). Despite the fact that post-training changes in the concentration of total phenolics in each group did not show statistically significant variations, the difference in post-training changes between the study groups was already statistically significant (*p* < 0.05).

It can be said that despite the increase in oxygen in the respiratory air in the study group during physical training, the therapeutic process did not lead to oxidative stress. It should be remembered that the aerobic mitochondrial metabolism on the respiratory chain is a source of a huge number of free radicals.

Physical exertion causes, in all individuals, a significant increase in oxygen consumption, exceeding up to 200 times that at the rest. Under these conditions, ROS production increases. An increase in markers depicting increased lipid peroxidation has been noted in obese patients after submaximal exercise [26,27,28,29]. In contrast, therapeutic oxygen therapy alone in patients with COPD resulted in a significant reduction in the severity of exercise-induced oxidative stress and inflammation process [15].

Our study shows that breathing an oxygen-enriched air mixture at a flow rate of 6 L/min during exercise results in less acidification of the body compared to the metabolic response of the subjects to exercise of the same load and breathing atmospheric air [30]. The effect of less metabolic acidosis is an inhibitory effect on the disturbance of the homeostasis of Ca^2+^ ions and, in turn, the activation of enzymes (calpains) which can lead to the disruption of oxidative phosphorylation on the mitochondrial chain [31]. As suggested by Radak et al., one can cite the hormesis theory described earlier, in which oxidative stress arising in the body is a stimulator of beneficial adaptive changes in the body and an increase in the body’s oxidative potential [32,33].

## 5. Conclusions

Increasing oxygen concentration in the breathing mixture under increased exercise as shown to be safe because it did not exacerbate oxidative stress. It is possible to develop a physiological model for the treatment of obesity by breathing an oxygen-enriched air mixture during a controlled individualized training program.

A limitation of this study is the absence of men who qualified for this experiment. Further studies should focus on this group.

## Figures and Tables

**Table 1 nutrients-15-00241-t001:** The effect of exercises with oxygen supplementation on the anthropometric characteristics (Me (Q1–Q3)) in obese patients.

	Study Group (*n* = 30)			Control Group (*n* = 30)		
Variable	First Research Period	Second Research Period	*p*-Value	First Research Period	Second Research Period	*p*-Value
Age (years)	43.00 (32.00–53.00)			49.00 (33.00–55.00)		
Body height (cm)	166.00 (160.0–170.0)			166.00 (164.00–168.00)		
Body weight (kg)	102.25 (95.00–122.40)	98.40 (92.90–117.30)	<0.0001	113.30 (97.60–134.80)	110.55 (93.30–125.00)	<0.0001
BMI (kg/m^2^)	40.00 (33.20–45.20)	37.90 (33.30–43.70)	<0.0001	40.63 (36.50–47.96)	39.85 (35.10–45.30)	<0.0001

*p*-value: Wilcoxon’s non-parametric paired-order test.

**Table 2 nutrients-15-00241-t002:** The effect of exercises with oxygen supplementation on plasma biochemical indicators and arterial blood gases (Me (Q1–Q3)) in obese patients.

	Study Group (*n* = 30)			Control Group (*n* = 30)		
Variable	First Research Period	Second Research Period	*p*-Value	First Research Period	Second Research Period	*p*-Value
CRP (mg/L)	4.60 (2.50–6.40)	3.70 (1.80–7.00)	0.1094	4.40 (2.50–6.30)	4.60 (1.40–6.30)	0.0750
FRAP (μmol/L)	1023.13 (882.48–1198.59)	952.15 (815.57–1104.85)	0.3447	1024.61 (903.80–1152.78)	1013.97 (838.62–1194.22)	0.3268
TBARS (μmol/L)	4.98 (4.12–6.29)	4.97 (4.10–5.81)	0.8501	5.03 (4.55–7.70)	4.74 (3.34–5.50)	0.0775
Total phenolics (g GAE/L)	2.48 (2.18–2.71)	2.44 (2.27–2.84)	0.0889	2.96 (2.74–3.07)	2.82 (2.72–2.99)	0.8445
pH	7.42 (7.39–7.42)	7.41 (7.39–7.41)	0.4140	7.41 (7.39–7.43)	7.41 (7.39–7.42)	0.0152
pCO_2_ (mmHg)	37.20 (34.60–40.20)	36.70 (35.00–38.05)	0.1828	36.75 (34.15–39.45)	36.10 (33.60–38.05)	0.4321
pO_2_ (mmHg)	70.30 (63.60–73.90)	73.75 (70.06–76.75)	0.0242	69.15 (65.20–74.00)	70.75 (67.45–74.90)	0.1982
HCO_3_^−^ [mmol/L]	23.00 (22.50–24.70)	23.00 (22.20–23.85)	0.4311	23.30 (22.45–24.30)	22.60 (21.40–23.55)	0.6319
BE [mmol/L]	−1.50 (−2.20–0.50)	−1.50 (−2.60–0.55)	0.3681	−1.25 (−2.35–0.05)	−2.05 (−3.60–−0.85)	0.0635
SpO_2_ [%]	97.00 (95.00–97.00)	97.00 (94.00–98.00)	0.3065	97.00 (96.00–98.00)	97.00 (95.00–98.00)	0.6319
La [mmol/L]	2.08 (1.22–2.93)	2.36 (1.23–3.40)	0.2845	1.94 (1.30–3.45)	2.47 (1.67–4.03)	0.0578

CRP: *C*-reactive protein; FRAP: ferric-reducing ability of plasma; TBARS: thiobarbituric acid reactive substances; pH: balance of acids and bases in the blood; HCO_3_^−^: bicarbonate; BE: base excess; La: lactate; *p*-value: Wilcoxon’s non-parametric paired-order test.

## Data Availability

The data presented in this study are available on request from the corresponding author. The data are not publicly available due to the consent provided by participants on the use of confidential data.

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
