# Peer review of "Assessment of Oxidative Stress Indices and Total Phenolics Concentrations in Obese Adult Women—The Effect of Training with Supplemental Oxygen: A Randomized Controlled Trial"

_nutrients, 2023, doi:10.3390/nu15010241_

Round 1

Reviewer 1 Report

Article:

Assessment of Oxidative Stress Indices and Total Phenolics concentration in Obese Patients—The Effect of Training with Supplemental Oxygen: A Randomized Controlled Trial

 An original article was made titled:  Assessment of Oxidative Stress Indices and Total Phenolics concentration in Obese Patients—The Effect of Training with Supplemental Oxygen: A Randomized Controlled Trial. The objective was to determine the effect of using an oxygen-enriched breathing mixture during controlled physical training on blood oxidative stress parameters and total phenolics (TP) concentration in obese patients.  The study investigated a novel, original and relevant topic. Comments and suggestions to strengthen the manuscript are presented below.

1.       Title: The title should indicate that only adult and aged women are included

2.       Abstract: In the abstract, it is suggested to indicate the study design

3.       Keywords: I suggest that the keywords be searched in the Mesh database (https://www.ncbi.nlm.nih.gov/mesh/)

4.       The introduction was clear. The current state of the research field was reviewed, and key publications cited, and the main aim of the work was mentioned. However, the research question or research hypothesis is not explicitly presented. I suggest adding at least one of them. Besides, I suggest indicating whether there are previous studies only in women or indicating whether gender is a factor that modifies the results of oxygen supplementation. these issues are relevant considering that the study was only carried out in adult women.

5.       Methods: They was described with enough detail to allow others to replicate and build on published results. I only suggest reviewing the exclusion criteria of the study; women should not do another exercise program at the same time. In addition, it must be indicated how the recruitment and randomization of the sample was done, as well as the sampling.

6.       Results: Provide a detailed and precise description of the results of study. In addition, it is organized according to the variables analyzed. it is very clear. only the results obtained between the groups are not presented in tables. I suggest presenting them, even if no differences were found between the groups.

7.       Discussion: Authors discussed the results and how they can be interpreted in perspective of previous studies. Although, little is said about their implications or limitations. I suggest adding a section that explicitly accounts for possible clinical implications and practical contributions of yours results

Author Response

Dear Reviewer.

Thank you for providing these insights. We wish to express our sincerest appreciation for your insightful comments on our paper, which have helped us significantly improve the quality of our paper. Below, we address each comment and indicate the location of changes, which are highlighted in yellow, in the revised manuscript.

Reviewer 2 Report

In the manuscript submitted to me for review entitled: „Assessment of Oxidative Stress Indices and Total Phenolics Concentration in Obese Patients—The Effect of Training with Supplemental Oxygen: A Randomized Controlled Trial the authors conducted a study that aimed to determine the effect of inhalation of an oxygen-enriched mixture during controlled exercise on parameters of blood oxidative stress and total phenolic content in obese patients.

The presented study is important because it may contribute to the development of a physiological model for controlling body weight, which is a problem for an increasing number of people.

My questions and remarks to the authors are:

1. On line 241 the following is presented: 47.00 (32.00-54.00). I personally did not understand what it means, if it can be clarified.

2. In Table 1, there is a row shift in the left column. Let them get better.

3. On line 96 is the place to enter the full meaning of the abbreviation BMI, which makes it superfluous after table 1 (but if you wish you can keep it after the table).

4. Under table 1 and 2, the meaning of the p-values must be given - from the comparison of which values the value was derived. Even in the Methods section point 2.6. Statistics does not specify what "p" is, and which statistical model it is.

5. I assume the authors have submitted an approval from the local institutional review board (IRB) or other appropriate ethics committee for conducting research involving humans.

6. In The References, literary source number 10 is without a given year. Let it be added.

Author Response

(The authors gave the same response as above.)

Round 2

Reviewer 1 Report

The authors have taken the comments into consideration and an improved version of the manuscript can now be read.